# Expression and Functional Role of the P2X7 Receptor in Glioma Models and Tissues: A Systematic Review

**DOI:** 10.3390/cancers17233748

**Published:** 2025-11-24

**Authors:** Maciej Baron, Piotr Lewandowski, Bartosz Bula, Kamil Liberka, Andrzej Skrzypiec, Marcin Fyrla, Maciej Czwakiel, Małgorzata Wachowicz, Romuald Wojnicz

**Affiliations:** 1Department of Pathomorphology, Faculty of Medical Sciences in Zabrze, Medical University of Silesia in Katowice, 40-055 Katowice, Poland; 2Department of Histology and Cell Pathology, Faculty of Medical Sciences in Zabrze, Medical University of Silesia in Katowice, 40-055 Katowice, Poland

**Keywords:** P2X7 receptor, glioma, glioblastoma, glioblastoma multiforme, astrocytoma, purinergic signalling, therapeutic target

## Abstract

Gliomas are the most common primary brain tumours, with glioblastoma multiforme (WHO grade IV) representing the most frequent and aggressive subtype. Despite advances in medicine, the prognosis for patients with glioblastoma remains poor, highlighting the urgent need for novel therapeutic strategies. In this study, we systematically reviewed the available literature to evaluate the role and potential clinical applications of the purinergic P2X7 receptor (P2X7R) in gliomas. A total of 41 studies were included and synthesized narratively. Our findings indicate that P2X7R is involved in multiple aspects of glioma biology, but its effects are context-dependent and sometimes contradictory. This review may help to guide future research directions and support the exploration of P2X7R as a potential therapeutic target in gliomas.

## 1. Introduction

Gliomas are the most common primary brain tumours in humans, and they are believed to originate from neuroglial progenitor cells [1]. According to the 2021 World Health Organization (WHO) classification, the most frequent diffuse gliomas in adults include astrocytoma IDH-mutant (grades 2, 3, or 4), oligodendroglioma IDH-mutant and 1p/19q co-deleted (grades 2 or 3), and glioblastoma (GBM) IDH-wildtype (grade 4) [2,3].

The histological structure of gliomas is highly heterogeneous. Low-grade gliomas are typically characterized by nuclear atypia and increased mitotic activity, while high-grade tumours also show areas of microvascular proliferation and necrosis [4]. GBM is not only the most common but also the most aggressive diffuse glioma [2,5]. Despite multimodal treatment—surgical resection followed by radiotherapy and chemotherapy with temozolomide—the prognosis remains poor, with a median survival of less than two years from diagnosis [6].

Therapeutic failure is attributed to several factors, including the infiltrative nature of gliomas, the presence of the blood–brain barrier, and the unique immunological environment of the central nervous system, all of which reduce the efficacy of conventional treatments [6]. These challenges underscore the urgent need for novel treatment strategies and molecular targets. A growing body of evidence suggests that the P2X7 receptor (P2X7R) may play a key role in glioma biology and could represent a promising therapeutic target.

The P2X7 receptor (P2X7R) is a transmembrane protein belonging to the family of ligand-gated ion channels. A functional P2X7R is formed by three subunits, each consisting of three principal domains: a large extracellular region that binds ATP, two transmembrane domains (TM1 and TM2) that line the ion channel, and intracellular N- and C-terminal regions. The C-terminal domain plays a crucial role in intracellular signal transduction by interacting with cytoplasmic proteins. A schematic representation of the P2X7R subunit and its molecular pathways is presented in Figure 1.

ATP is the natural ligand of P2X7R. Interestingly, the effect of receptor stimulation depends on ATP concentration and duration of exposure. Low ATP concentrations or short stimulation times lead to the opening of a cation-selective ion channel, permitting Na^+^ and Ca^2+^ influx and K^+^ efflux. In contrast, high ATP concentrations or prolonged stimulation result in the formation of large, non-selective membrane pores that allow the passage of molecules up to 900 Da, including fluorescent dyes such as YO-PRO-1 and ethidium bromide [7].

Beyond its role as an ion channel, P2X7R also participates in inflammatory and immune signalling by activating several downstream pathways, including NF-κB and VEGF (vascular endothelial growth factor). It is expressed in a variety of tumours and has been implicated in processes such as proliferation and cell migration [7]. Consequently, P2X7R is increasingly considered a promising therapeutic target in oncology, including in the context of gliomas.

## 2. Materials and Methods

The aim of this review was to explore the expression, functional role, and potential therapeutic implications of the P2X7 receptor in gliomas. A systematic literature search was conducted in three databases: PubMed, Scopus, and Web of Science, completed on 12 April 2025. The following search terms were used: (“P2X7” OR “P2X7R” OR “P2X7 receptor”) AND (“glioblastoma” OR “GBM” OR “glioma”). Search results were managed and initially screened using Zotero. This review was registered in PROSPERO (CRD420251112608) [8].

After merging the search results from all three databases, duplicate records were identified and removed using both automated filters and manual inspection in Microsoft Excel. Following deduplication, 95 unique studies were retained for screening. Each study was independently screened by title and abstract by two reviewers (M.B. and B.B.), with discrepancies resolved through discussion and consensus. Studies deemed relevant were selected for full-text review. Ultimately, 41 articles met the inclusion criteria and were included in the final analysis. Data extraction was performed independently by three reviewers using a standardized data collection sheet. Any disagreements were resolved by consensus. No automation tools were used in the data collection process. Detailed inclusion and exclusion criteria are presented in Appendix A. To evaluate methodological quality and risk of bias, the OHAT Risk of Bias Tool was applied [9]. Results of the risk of bias assessment are summarized in Appendix A. No formal assessment of the overall certainty of evidence (e.g., using GRADE) was performed due to the qualitative nature of the synthesis.

In addition to the main search, a supplementary search was conducted in Embase using the same search terms. Most retrieved records were duplicates of studies already identified in the primary databases. A small number of unique records were identified; however, all were excluded as they did not meet the predefined inclusion criteria.

Data from the included studies were manually extracted into a structured Excel spreadsheet. Extracted variables included glioma model, type of P2X7R modulation (activation or inhibition), downstream molecular effects, and primary outcomes. Based on these results, each study was assigned to one or more thematic categories, including receptor expression, effects on proliferation, migration, immune regulation, receptor functionality, and therapeutic interactions. In one case, data were extracted exclusively from the abstract due to lack of full-text availability; however, the abstract provided sufficient details on experimental models and outcomes to meet the predefined inclusion criteria [10].

Due to substantial heterogeneity in study designs, glioma models, and outcome measures, no meta-analysis was performed. Instead, a structured narrative synthesis was applied, supported by summary tables organized by cell line and experimental outcomes. This review was conducted in accordance with PRISMA (Preferred Reporting Items for Systematic Reviews and Meta-Analyses) guidelines. A detailed PRISMA flowchart is presented in Figure 2, and the PRISMA 2020 checklist is available in Appendix A [11].

## 3. Results

The database search yielded 95 unique records after deduplication. Following title and abstract screening, as well as full-text assessment, 54 studies were excluded as irrelevant. Ultimately, 41 articles met the eligibility criteria and were included in the qualitative synthesis. The selection process is illustrated in the PRISMA flowchart (Figure 2). Of the 41 included studies, 39 employed in vitro glioma models, 9 used in vivo xenograft models, and 17 analyzed patient-derived tissues or glioma stem cells. Several studies used more than one experimental model; therefore, the numbers across categories exceed the total number of included studies

### 3.1. P2X7R Expression in Gliomas

A total of 23 studies have reported the expression of the P2X7 receptor in glioma tissues and cell models. The receptor is detected in various glioma cell lines (e.g., U87, U251, M059J/K, C6, GL261), patient-derived glioma samples, glioma stem cells (GSCs), and even tumour-associated microglia [10,12,13,14,15,16,17,18,19,20,21,22,23,24,25,26,27,28,29,30]. However, the expression levels of P2X7R vary considerably between studies, depending on tumour grade, cell type, and environmental conditions [12,14,18,19,20,25,31]. These findings suggest that P2X7R expression in gliomas is both dynamic and context dependent.

Notably, P2X7R expression appears to be influenced by environmental factors. For instance, hypoxia, retinoic acid, oxidative stress, and radiotherapy have all been reported to modulate P2X7R levels. In some glioma cell lines, such as M059J/K or A172, exposure to retinoic acid or irradiation resulted in upregulation of P2X7R [22,32,33].

The included studies revealed contradictory findings regarding tumour grade. Three studies reported an increase in P2X7R expression with advancing tumour grade and stage IV glioma [18,19,20], whereas three others suggested the opposite—lower P2X7R expression correlating with stage IV glioma [12,14,31]. Interestingly, one study found elevated P2X7R expression in stage II and III gliomas compared with normal brain tissue and stage IV glioma [12]. Taken together, these findings suggest that the role of P2X7R may be dualistic, varying depending on the biological state of glioma cells and tumour microenvironmental factors.

In contrast, one older study using the C6Bu-1 subline reported a lack of detectable P2X7R expression in these cells, suggesting potential subline-specific differences or technical limitations [34].

### 3.2. Influence of P2X7R Modulation for Proliferation and Survival of Glioma Cells

The purinergic P2X7 receptor has been implicated in the regulation of both proliferation and cell death in various tumour types, including gliomas. This section reviews studies investigating how modulation of P2X7R affects glioma cell growth and viability.

A total of 19 studies addressing this issue were identified. These studies employed various glioma cell models, including C6, U138, GL261, U251, TS15-88, A172, M059J, M059K, U138-MG, U87, and patient-derived glioma cells. In most studies, modulation of P2X7R function affected tumour cell proliferation or survival, although the nature of this effect varied.

In three models, P2X7R activation (e.g., by eATP or BzATP) promoted glioma cell proliferation, for example, in U87, U251, and U138 cell lines, as well as in primary cells from patients [12,19,35]. However, in two cases, particularly in glioma stem cells, P2X7R activation resulted in decreased proliferation or even induction of cell death [26,27]. Moreover, two studies revealed that high concentrations of P2X7R agonists (specifically BzATP or ATP at around 5 mM) can exert cytotoxic effects on glioma cells [27,28]. On the other hand, one study reported that glioma cell lines (U138-MG and C6) showed significant resistance to extracellular ATP compared with control hippocampal cultures [36].

Conversely, eight studies showed that inhibition of P2X7R by specific antagonists (e.g., BBG, AZ10606120, A740003, oATP) reduced proliferation, glioma cell numbers, and tumour mass [12,15,20,21,24,30,37,38]. One additional study suggested that increased P2X7R expression decreased glioma cell survival [32]. Interestingly, another study reported that inhibition of P2X7R by BBG paradoxically increased proliferation [39].

The impact of P2X7R on proliferation also extended to downstream apoptotic signalling. One study found that stimulation with BzATP reduced caspase-3 and caspase-8 activation [18], while two others reported that P2X7R inhibition increased resistance to the cytotoxic effects of NAD^+^ and ATP [22,40].

These discrepancies likely arise from intrinsic differences between glioma cell lines, experimental conditions (e.g., ATP concentration, exposure duration), or the presence of different P2X7R isoforms or splice variants. This highlights the functional complexity of P2X7R signalling in glioma biology.

### 3.3. P2X7R Functionality in Gliomas

The P2X7 receptor exhibits preserved functionality as both an ion channel and a pore-forming receptor in multiple glioma models. Its activity has been demonstrated in at least five glioma systems (C6, U251, GL261, and GSCs) as well as in glioma-associated microglia, through calcium influx assays and dye uptake (ethidium bromide, YO-PRO-1). These responses were consistently blocked by selective antagonists (AZ10606120, oATP, A740003), confirming receptor specificity [12,17,21,23,28,30,33].

P2X7R activation initiates multiple intracellular signalling cascades involved in cell proliferation, survival, and migration. It induces phosphorylation of several kinases, including p38 MAPK, Akt, ERK1/2, MEK1/2, Pyk2, SMAD2, and PKCδ [12,18,19,22,41]. These pathways contribute to increased expression of PCNA, enhanced migratory potential, epithelial–mesenchymal transition (EMT), and greater cellular invasiveness [18,19,42,43]. Additionally, P2X7R interacts with other signalling pathways such as TGFβ. Activation of P2X7R and TGFβ signalling in glioma stem cells has been shown to promote EMT [18], while in one study (C6 cell line), inhibition of P2X7R paradoxically increased EGFR activity and angiogenesis [39]. Interestingly, P2X7R inhibition has also been shown to impair the recruitment of endothelial progenitor cells to glioma tissue, suggesting a role in tumour-associated angiogenesis [44].

Another important function of P2X7R in glioma is its role in intercellular communication through the regulation of extracellular vesicle (EV) secretion. In glioma stem cells, P2X7R activation alters the protein composition of EVs, increasing the abundance of proteins involved in cell motility, oxidative stress response, chromatin remodelling, and transcriptional regulation [42].

P2X7R activity also intersects with NAD^+^ signalling and redox balance: NAD^+^ exposure reduced glioma viability in a P2X7R- and calcium-dependent manner, an effect abolished by P2X7R antagonists [40]. This indicates that P2X7R contributes to glioma vulnerability under metabolic stress.

Nevertheless, receptor functionality appears context-dependent. In two studies using the C6 line, canonical P2X7R responses were absent and instead mediated by TRPM7 or P2Y2 channels [45,46].

### 3.4. P2X7R Influence on Glioma Immunology

P2X7R may play a role in shaping the immune response within glioma tissue. It is abundantly expressed on microglia and macrophages infiltrating gliomas, both in human samples and animal models [10,20,30,47]. Inhibition of P2X7R has been shown to reduce the number of microglial cells within glioma tissue [30,47].

Additionally, P2X7R modulation—either activation by ATP or BzATP, or inhibition by antagonists such as A740003 or BBG—influences the expression of multiple immune mediators, predominantly pro-inflammatory cytokines and chemokines, including MCP-1 (CCL2), IL-8 (CXCL8), VEGF, MIP-1α, IL-2, IL-6, IL-10 [12,23,47,48,49,50]. Many of these factors are secreted by glioma-associated microglia and macrophages (GAMs), although glioma cells themselves may also contribute to the inflammatory milieu [47,49,50].

Notably, stimulation of glioma cells (e.g., C6 or U251) has been associated with increased expression of IL-8, MCP-1, and VEGF, suggesting that P2X7R directly influences glioma cell-driven immune signalling [23,50].

Moreover, glioma cells may modulate microglial function through the release of GM-CSF, which promotes microglial ATP secretion-activating P2X7R in an autocrine or paracrine manner. Inhibition of P2X7R with AZ10606120 reduced GM-CSF expression, indicating a potential feedback loop between P2X7R signalling and glioma-microglia interaction [21,35]. Given the heterogeneity of in vitro models and experimental designs, the most consistent findings are summarized in Table 1.

### 3.5. P2X7R and Invasiveness

One of the hallmark features of malignant tumours is their ability to invade surrounding tissues. Emerging evidence suggests that the P2X7 receptor may contribute to the invasive phenotype of glioma cells by modulating migratory capacity, although findings remain partially inconsistent.

Two studies reported that activation of P2X7R by the selective agonist BzATP enhanced glioma cell migration in scratch wound assays [18,19]. In contrast, another study using the same assay observed that P2X7R activation reduced migratory potential [32]. Furthermore, knockdown of P2X7R expression in GL261 cells was associated with decreased adhesion but increased migratory activity [27]. These discrepancies may reflect intrinsic differences between cell types, experimental models, or assay conditions.

Notably, P2X7R signalling has also been linked to epithelial–mesenchymal transition (EMT) in glioma stem cells. Stimulation with BzATP upregulated mesenchymal markers, including N-cadherin, ZEB1, Snail1, Twist1, and vimentin. This effect was reversed by the selective antagonist A438079, indicating a direct role of P2X7R in promoting an invasive molecular phenotype [18].

### 3.6. Evidence from Human-Derived Tissues

Although gliomas are highly heterogeneous tumours, data obtained from patient-derived tissues provide particularly valuable insights that may better reflect clinical relevance. This section summarizes the evidence from studies performed directly on human glioma tissues, GSCs, and imaging data.

P2X7R is expressed in human glioma tissue, including in tumour cells, glioma-associated microglia, and GSCs [10,12,19,20,24,25,26,30,35]. Notably, human gliomas may express distinct P2X7R isoforms, namely the canonical pore-forming P2X7R-A and the truncated non-pore-forming P2X7R-B. Radiotherapy has been reported to increase cellular cholesterol levels, which may inhibit P2X7R-A macropore formation while supporting the signalling function of P2X7R-B, potentially contributing to treatment resistance [51]. Interestingly, one clinical study found that higher P2X7R expression in glioma tissue was associated with better response to radiotherapy and improved overall survival [14]. In vitro data using patient-derived samples also indicate that hypoxia can upregulate P2X7R expression [25].

However, findings regarding the correlation between P2X7R expression and glioma grade are inconsistent. Two studies report increased P2X7R expression with higher WHO tumour grade (II–IV) [19,20], while two other studies found reduced expression in grade IV tumours, potentially due to P2X7R promoter hypermethylation [12,31].

Functional studies on GSCs have demonstrated that stimulation with the P2X7R agonist BzATP increases the expression of epithelial–mesenchymal transition (EMT)-related proteins, including N-cadherin, ZEB1, Snail1, Twist1, and vimentin. This effect was reversed by the P2X7R antagonist A438079 [18]. BzATP also enhanced the migratory and invasive capacities of GSCs in scratch wound and transwell assays, respectively [18]. Additionally, P2X7R stimulation increased the release of extracellular vesicles enriched in proteins associated with cytoskeletal remodelling, energy metabolism, and stress responses (e.g., vimentin, actin, ARP3) [42]. Moreover, BzATP treatment reduced apoptosis in GSCs by downregulating caspase 3 and 7 levels [18].

The impact of P2X7R modulation on glioma cell proliferation and survival in human-derived models remains inconclusive. One study indicates that BzATP stimulation reduces proliferation of GSCs, and co-treatment with temozolomide (TMZ) further enhances cytotoxicity [26]. In contrast, another study shows that P2X7R antagonists such as BBG reduce glioma cell numbers [30]. The most relevant findings from studies performed on patient-derived glioma tissues and glioma stem cells are summarized in Table 2. These data highlight the complex and context-dependent role of P2X7R in glioma biology, emphasizing the need for further research to clarify its prognostic and therapeutic significance.

### 3.7. P2X7R and Modern Therapies

Radiotherapy remains one of the standard treatment strategies for gliomas. Three studies suggest that modulation of the P2X7 receptor may influence the cellular response to ionizing radiation. However, the precise effect of P2X7R activation or inhibition remains unclear.

There is evidence that P2X7R blockade with selective antagonists (e.g., AZ10606120, A740003) can enhance the cytotoxic effect of radiotherapy, increasing glioma cell death. Post-irradiation administration of P2X7R antagonists has been associated with increased apoptosis and reduced cell viability [51].

Conversely, the other two studies indicate that P2X7R activation following radiation exposure, through extracellular ATP accumulation, also contributes to apoptosis. Co-treatment with ATP and irradiation increases annexin V binding and bromide dye uptake, suggesting a synergistic pro-death effect of ATP and radiation. This effect was abolished by the selective P2X7R antagonist A740003 [14,33]. The discrepancies in response may be partially explained by isoform-specific expression changes following irradiation—P2X7A (pore-forming and pro-apoptotic) is downregulated, while P2X7B (non-pore-forming and pro-survival) becomes dominant in some glioma populations [51].

Another therapeutic avenue is pharmacological treatment. The alkylating agent temozolomide (TMZ) is the current first-line chemotherapy for glioma. Interestingly, P2X7R antagonism with AZ10606120 has been shown to be more effective than TMZ alone in reducing glioma cell numbers in U251 cells, although no significant synergistic effect was observed when both agents were combined [20,21,37]. A similar pattern was observed in glioma stem cells, with P2X7R antagonists outperforming TMZ and no synergy detected in combination treatments [17].

However, P2X7R activation may enhance TMZ efficacy via a different mechanism. As P2X7R activation leads to membrane pore formation, it may facilitate increased intracellular entry of TMZ. Studies using BzATP (a P2X7R agonist) in combination with TMZ demonstrated enhanced cytotoxicity and reduced migration, particularly in patient-derived glioma stem cells and differentiated A172 cells [26,32].

### 3.8. Risk of Bias Assessment

Overall, most included studies were rated as low risk of bias. Domains most frequently rated as “No” or “Not applicable” included randomization, allocation concealment, and blinding, which were rarely applied in in vitro studies. Conversely, exposure characterization and outcome assessment were generally well described, and selective reporting was uncommon. A small number of studies raised concerns due to incomplete data or reliance on abstract-only reports. Full details of the risk of bias assessment are summarized in Appendix A.

## 4. Discussion

Gliomas remain among the most therapeutically challenging malignancies, characterized by aggressive growth, diffuse infiltration, and resistance to standard therapies. Therefore, identifying novel molecular targets is essential. The P2X7 receptor (P2X7R), due to its involvement in cell signalling, inflammation, and regulation of cell fate, has emerged as a potential candidate.

However, results from the studies included in this review demonstrate that the effects of P2X7R modulation are not uniform. This heterogeneity likely reflects both the intrinsic diversity of glioma subtypes and differences in experimental conditions. For instance, in U251 glioma cells, P2X7R activation has been shown to promote proliferation, whereas in GSCs, the same activation can induce cytotoxic effects or reduce proliferation. Such contrasting outcomes suggest that P2X7R signalling may be highly context-dependent.

Despite these discrepancies, the majority of reviewed studies reported that P2X7R modulation-whether through agonists or antagonists-produces significant effects on glioma cell behaviour, including proliferation, migration, cytokine secretion, and survival. This supports the hypothesis that P2X7R is functionally active and relevant in glioma biology.

Further research is needed to clarify the role of P2X7R isoforms, the influence of the tumour microenvironment, and the impact of combinatorial treatments (e.g., with temozolomide or radiotherapy). A deeper mechanistic understanding will be crucial to determine whether P2X7R can be effectively and safely targeted in clinical settings.

P2X7R also appears to play an important role in shaping the immune landscape of gliomas. It is expressed on tumour-associated microglia and macrophages, and its pharmacological inhibition has been shown to reduce the levels of these immune cells within the tumour microenvironment. Furthermore, P2X7R modulation leads to changes in the expression of various cytokines by both glioma cells and infiltrating immune cells, indicating its involvement in pro-inflammatory signalling pathways. Figure 3 summarizes the proposed role of P2X7R across major hallmarks of cancer, including proliferation, resistance to apoptosis, invasion, and tumour-promoting inflammation.

Interestingly, P2X7R modulation may also hold therapeutic relevance. Several studies suggest that altering P2X7R activity can enhance the effects of radiotherapy; however, the data are inconsistent regarding whether receptor activation or inhibition yields the most beneficial outcomes. These discrepancies may be attributable to differences in glioma models, experimental conditions, or specific cell populations investigated. Moreover, due to its role in forming membrane pores, P2X7R activation may facilitate intracellular entry of temozolomide and thereby increase its cytotoxicity against glioma cells.

It is important to note that the effects of P2X7R modulation are highly context dependent. Differences in experimental design, cell line characteristics, and the inherent heterogeneity of glioma subpopulations all contribute to the variability of observed outcomes. These factors underscore the need for further investigation using well-standardized models that better reflect the complexity of human gliomas.

Although the current evidence remains partially inconsistent, these findings may reflect the intrinsic heterogeneity of gliomas rather than methodological limitations. It is conceivable that P2X7R serves as a molecular determinant of tumour behaviour in specific glioma subtypes. Similar to HER2 in breast cancer, P2X7R could emerge as a targetable biomarker for selected patients exhibiting high receptor expression and preserved functional activity. Further in vivo and translational studies are warranted to verify this concept. Notably, several selective P2X7R antagonists, such as A740003 and GSK1482160, are known to cross the blood–brain barrier, supporting the feasibility of future in vivo investigations.

Taken together, the current evidence suggests that P2X7R represents a biologically meaningful and pharmacologically actionable target in glioma. However, further preclinical and clinical studies are necessary to clarify its role and therapeutic potential in glioma treatment.

### 4.1. Study Limitations

This review has several limitations. Due to the marked heterogeneity of glioma models, experimental designs, and outcome measures, no meta-analysis could be performed. Most included studies relied on in vitro glioma models with limited in vivo validation, which restricts the generalizability of the findings to clinical settings. In one case, data were extracted from the abstract only, as full text was unavailable, which may have limited methodological assessment. Additionally, variability in the methods used to evaluate P2X7R expression and the selectivity of pharmacological ligands complicates cross-study comparisons. Finally, potential publication bias cannot be excluded, as negative results are less likely to be published.

### 4.2. Recommendations for Future Studies

P2X7R represents a promising target for glioma therapy; however, the considerable variability in findings across studies raises important questions about its clinical applicability. Future research should prioritize methodological rigour, including standardized experimental designs, consistent outcome reporting, and careful control of confounding factors. While in vitro models provide valuable mechanistic insights, their results often diverge from in vivo and patient-derived systems. Therefore, more emphasis should be placed on translational studies, particularly those utilizing patient tissues and glioma stem cells, to better reflect the heterogeneity of the human disease.

## 5. Conclusions

Despite some discrepancies among the available data, the P2X7 receptor emerges as a promising target with potential clinical applications in gliomas. Evidence synthesized in this review highlights its involvement in key pathophysiological processes, including cell proliferation, survival, invasion, immune modulation, and cytokine secretion. However, the functional role of P2X7R appears to be highly context-dependent, varying across glioma models and patient-derived samples. Taken together, current findings underscore the need for well-designed preclinical and clinical studies to clarify the therapeutic relevance of P2X7R and its potential as a biomarker or target for glioma treatment.

## Figures and Tables

**Figure 1 cancers-17-03748-f001:**
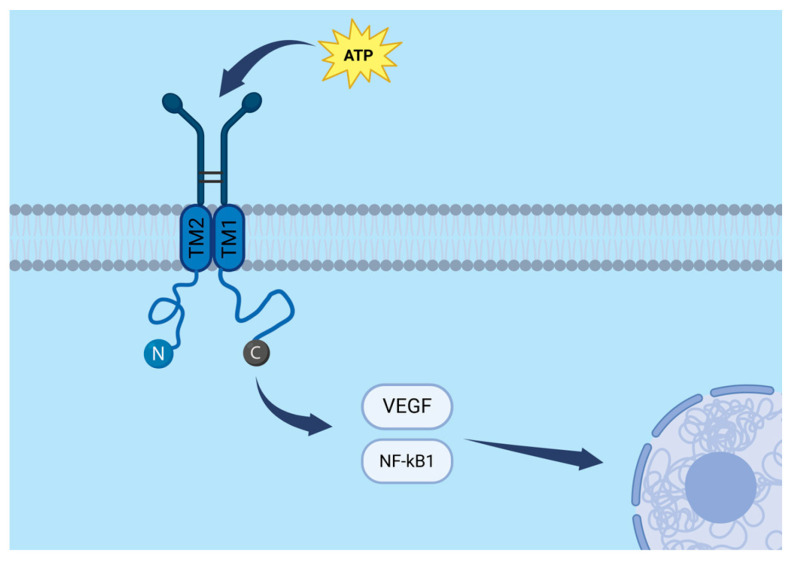
Molecular pathways and structural organization of a single P2X7R subunit, including the extracellular ATP-binding domain, two transmembrane segments (TM1 and TM2), and intracellular N- and C-terminal regions. The VEGF and NF-κB signalling pathways are schematically highlighted. *Created*
*in BioRender. Baron, M. (2025) https://BioRender.com/1z3sx7n.*

**Figure 2 cancers-17-03748-f002:**
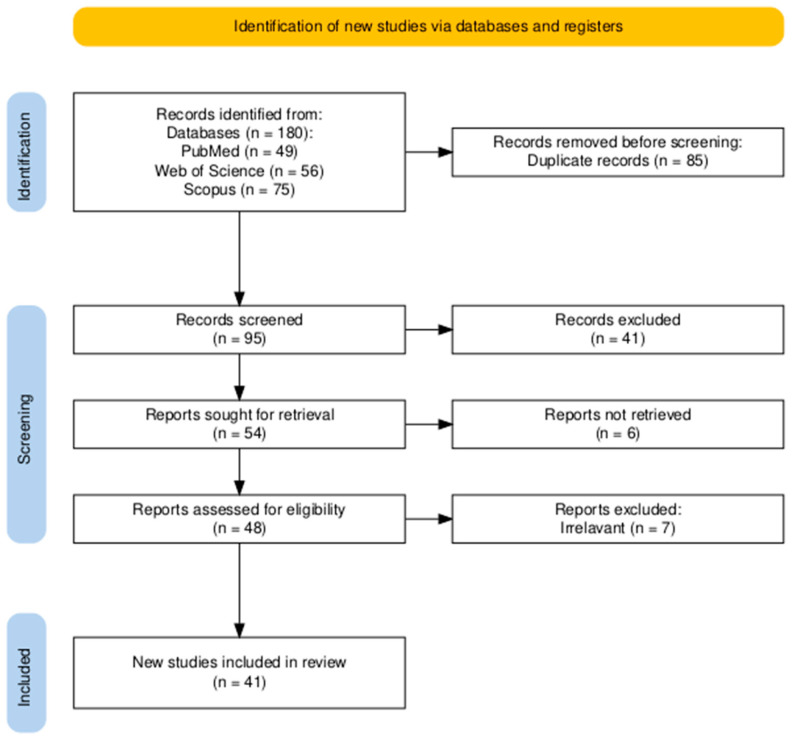
PRISMA 2020 flow diagram summarizing the selection process. After removal of duplicates, 95 records were screened by title and abstract. Fifty-four studies were excluded, and 41 full-text articles met the predefined eligibility criteria and were included in the qualitative synthesis.

**Figure 3 cancers-17-03748-f003:**
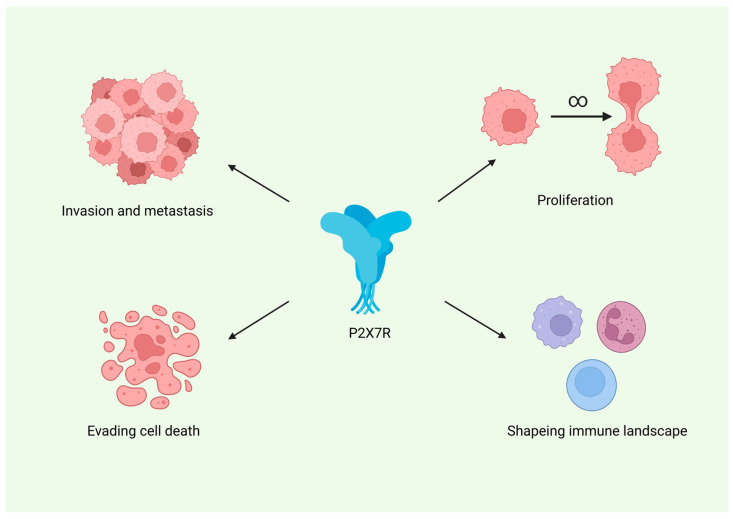
Proposed involvement of the P2X7 receptor (P2X7R) in key hallmarks of glioma progression. *Created in BioRender. Baron, M. (2026) https://BioRender.com/gv0mkrq.*

**Table 1 cancers-17-03748-t001:** Summary of representative in vitro findings on P2X7R modulation in glioma models, highlighting the diversity of experimental conditions and outcomes. P2X7R and invasiveness.

Cell Model	Experimental Intervention	Observed Effect	References
M059J/K, A172	Environmental modulation (hypoxia, retinoic acid, oxidative stress, radiotherapy)	Increased P2X7R expression levels.	[22,32,33]
U87, U251, U138	P2X7R activation (ATP, BzATP)	Promotion of cell proliferation	[12,19,35]
Glioma stem cells (GSCs)	P2X7R activation	Decreased proliferation and induced apoptosis in GSCs.	[26,27]
C6, U251	P2X7R activation (BzATP)	Induced release of pro-inflammatory cytokines and growth factors (IL-8, MCP-1, VEGF).	[23,50]

**Table 2 cancers-17-03748-t002:** Summary of key findings from studies performed on human-derived glioma tissues and glioma stem cells (GSCs).

Role/Process	Observed Effect or Finding	References
Expression pattern	P2X7R is expressed in glioma cells, glioma-associated microglia, and GSCs. Two main isoforms were identified: the pore-forming P2X7R-A and the truncated, non-pore-forming P2X7R-B.	[10,12,19,20,24,25,26,30,35]
Clinical outcomes	Higher P2X7R expression was linked to improved radiotherapy response and overall survival. However, radiotherapy may also promote a shift toward the resistant P2X7R-B isoform.	[14,51]
Correlation with tumour grade	Findings are inconsistent: some studies reported increased P2X7R expression with higher WHO grade (II–IV), while others observed reduced expression in grade IV tumours, possibly due to promoter hypermethylation.	[12,19,20,31]
Proliferation and therapy response	Contradictory effects were observed: BzATP reduced proliferation and enhanced TMZ cytotoxicity in GSCs, whereas P2X7R inhibition decreased glioma cell numbers.	[26,30]

## Data Availability

No new data was created or analyzed in this study. Data sharing is not applicable to this article.

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
