# Peer review of "Expression and Functional Role of the P2X7 Receptor in Glioma Models and Tissues: A Systematic Review"

_cancers, 2025, doi:10.3390/cancers17233748_

Round 1
Reviewer 1 Report
Comments and Suggestions for Authors
The presented review is discussing the role of purinergic 28 receptor P2X7 (P2X7R) in glioma pathogenesis and progression. The authors summarized the data on P2X7 effects in cancer cell signaling, studied in vitro and in vivo, and proposed P2X7 as potential target for pharmacological inhibition followed by the decrease of cancer cell viability, and for the development of novel therapies based on P2X7 inhibition. The review seems to be relevant to the scope of the Cancers Journal, and the amount of the reviewed data is sufficient for the publication. However, there several issues to be addressed:
- The review is lacking figures and tables. Figures illustrating the P2X7 role in discussed oncogenic signaling and tables summarizing and categorizing the findings in vitro and in vivo on P2X7 function should be added in the paper.
- Abbreviation list is missing.
- The English editing should be performed. Typos and grammar issues should be eliminated.
- Journal format requirements should be obeyed.
- It seems useful to check the role of P2X7 in each hallmark of cancer and add the corresponding literature data with summarizing Scheme or Figure.
- The hierarchy of the review should be more concise: authors should review the role of P2X7 in cancer cell signaling (including data from patients) and then move to the possible therapies.
The English editing should be performed. Typos and grammar issues should be eliminated.
Author Response
Thank you for your review we have made the corrections suggested by you:
Comments 1: [The review is lacking figures and tables. Figures illustrating the P2X7 role in discussed oncogenic signaling and tables summarizing and categorizing the findings in vitro and in vivo on P2X7 function should be added in the paper.]
Response 1: A new Figure 3 has been added to the Discussion section, illustrating the key roles of P2X7R in oncogenic signalling. In addition, summary tables presenting in vitro and human-derived data were included to enhance the clarity of results interpretation.
Comments 2: [Abbreviation list is missing].
Response 2: An Abbreviations list has been added (page 13).
Comments 3: [The English editing should be performed. Typos and grammar issues should be eliminated.]
Response 3: The manuscript has undergone thorough English language editing to eliminate typos and grammar issues. The language has been standardized to British English.
Comments 4: [Journal format requirements should be obeyed.]
Response 4: The manuscript has been reformatted according to Cancers journal guidelines, including figure and table captions, section hierarchy, and reference style.
Comments 5: [It seems useful to check the role of P2X7 in each hallmark of cancer and add the corresponding literature data with summarizing Scheme or Figure.]
Response 5: Figure 3 (page 12) was added to summarise the role of P2X7R in glioma pathobiology in the context of cancer hallmarks.
Comments 6: [The hierarchy of the review should be more concise: authors should review the role of P2X7 in cancer cell signaling (including data from patients) and then move to the possible therapies.]
Response 6: The section hierarchy has been reorganised. The subsection “P2X7R and modern therapies” (previously 3.6, now 3.7) was moved after “Evidence from human-derived tissues” to improve logical flow and conciseness.
Reviewer 2 Report
Comments and Suggestions for Authors
This manuscript presents a systematic review of the current literature evaluating the expression, function, and therapeutic implications of the purinergic P2X7 receptor (P2X7R) in gliomas. The topic is highly relevant and timely, as glioblastoma multiforme (GBM) remains one of the most aggressive and treatment-resistant brain tumors, and novel molecular targets are urgently needed.
The authors have followed a methodologically sound and transparent approach, performing a comprehensive literature search across major scientific databases and assessing the quality of included studies using the OHAT Risk of Bias Tool. The manuscript is well-structured, clearly written, and appropriately contextualized, making it a valuable contribution to the field of neuro-oncology and molecular pharmacology.
Having said the above-mentioned, I have not detected major flaws. I recommend it for publication.
Minor remarks:
Lines 59; 85: please provide citations
Please pay attention to provide full names of abbreviations the first time they appear in the text e.g. lines 87-88.
Author Response
Thank you very much for your positive feedback and helpful suggestions.
We have implemented the recommended changes:
Comments 1: [Lines 59; 85: please provide citations]
Response 1: Appropriate citations have been added in lines 59 and 85.
Comments 2: [Please pay attention to provide full names of abbreviations the first time they appear in the text e.g. lines 87-88.]
Response 2: Full names of abbreviations have been provided at their first appearance in the text.
Reviewer 3 Report
Comments and Suggestions for Authors
In this systematic review of the literature, the authors analyzed the supportive evidence for the involvement of purinergic P2X7 receptor (P2X7R) in glioma pathology. The authors analyzed the current data available on P2X7R specifically pertaining its expression levels, functionality and role in tumor progression, and potential clinical applications in gliomas. P2X7R is purinergic receptor that is widely distributed in the CNS and functions as a ligand-gated cation channel that is operated by ATP binding. The receptor primarily operates as an ion channel when activated at low ATP concentrations. At higher ATP concentrations, the activation of the receptor by leads to the recruitment and assembly of pannexin pores, which allow the trafficking of solutes (including ATP) across the plasma membrane. The receptor was found to be overexpressed by gliomas and is believed to play a pathological role in glioma progression. However, whether the expression of the receptor correlates with glioma grade is not entirely clear and also the role played by this receptor in glioma pathology is far from being well understood. Nonetheless, the authors found 41 studies in support of the notion that the overexpression and/or pharmacological modulation of P2X7R has various impacts on glioma cell behavior including effects on cell proliferation, survival, invasive properties, immune signaling, and response to treatment. While some of the reported findings appear to be contradictory, the authors conclude that P2X7R may nonetheless represent a biologically relevant and pharmacologically actionable target in gliomas with further studies needed to dissect its effects and confirm its clinical potential.
While the authors did follow a sound methodological approach for their study, I found the reported evidence supporting the biological contribution of P2X7R in gliomas to be rather confusing, with very little clarity when it comes to understating the contextual effects elicited by the pharmacological modulation of this receptor in glioma models. In fact, the vast majority of studies reviewed by the authors report rather contradictory in vitro observations, which are unfortunately quite hard to appraise in the absence of complementary in vivo data generated with pharmacological inhibitors in animal models of intracranial glioma. Some of these inhibitors are available and known to penetrate the blood-brain barrier. In the absence of such in vivo models, there is very little evidence (if any) to suggest that the pharmacological inhibition of P2X7R may lead to improvements in median survival in glioma. While I appreciate the effort made by the authors to systematically review the role of P2X7R in gliomas, I found the current evidence for this biological target rather lacking and inconsistent which makes me question the utility of their review.
Author Response
Comments 1: [While the authors did follow a sound methodological approach for their study, I found the reported evidence supporting the biological contribution of P2X7R in gliomas to be rather confusing, with very little clarity when it comes to understating the contextual effects elicited by the pharmacological modulation of this receptor in glioma models. In fact, the vast majority of studies reviewed by the authors report rather contradictory in vitro observations, which are unfortunately quite hard to appraise in the absence of complementary in vivo data generated with pharmacological inhibitors in animal models of intracranial glioma. Some of these inhibitors are available and known to penetrate the blood-brain barrier. In the absence of such in vivo models, there is very little evidence (if any) to suggest that the pharmacological inhibition of P2X7R may lead to improvements in median survival in glioma. While I appreciate the effort made by the authors to systematically review the role of P2X7R in gliomas, I found the current evidence for this biological target rather lacking and inconsistent which makes me question the utility of their review.]
Response 1:
Thank you for your valuable comments.
Although gliomas are among the most common primary brain tumours, they remain relatively rare overall. Moreover, the functional role of P2X7R in gliomas represents a relatively new and emerging research area. For these reasons, current evidence is largely based on in vitro studies, while in vivo investigations are still limited. These limitations, as well as the heterogeneity of available data, are explicitly discussed in the Limitations and Discussion sections.
Nevertheless, in the majority of studies reviewed, P2X7R modulation exerted a notable impact-whether promoting or inhibiting-on glioma cell behaviour, including proliferation, migration, and survival. Therefore, we believe that P2X7R holds potential as a promising therapeutic target-possibly analogous to HER2 in breast cancer-for selected glioma subtypes exhibiting appropriate receptor expression and/or functionality.
To address this point more clearly, we have added a paragraph in the Discussion section (page 13, lines 427–435) to emphasize this concept and the need for further in vivo validation.
Reviewer 4 Report
Comments and Suggestions for Authors
I thank the editors for the opportunity to review the article by Maciej Baron et al.
Abstract
The abstract omitted a results section. I suggest adding it.
Introduction
The introduction is short, which introduces the reader to the problems and issues related to gliomas. This review was registered in PROSPERO (CRD420251112608). Figure 1 is quite "provisional." It doesn't add anything new, so it suggests removing or expanding upon the pathways in which this receptor participates.
Materials and Methods
I have no reservations about the presented methodology. The method for searching for articles is understandable and can be repeated. However, I suggest improving the quality of Figure 2.
Results
The results section is well-organized and divided into subsections. The works cited in this section are clearly and appropriately summarized.
Discussion do not raise any objections.
Conclusions
The conclusions drawn from the review are sound. However, I'm curious if the authors plan to empirically analyze the functional role of the P2X7 receptor in glioma, either in animal models, cells, or tissues?
Thank you
Author Response
Thank you very much for your constructive feedback and helpful suggestions.
We have implemented all recommended changes:
Comments 1: [The abstract omitted a results section. I suggest adding it.]
Response 2: A results section has been explicitly designated within the abstract
Comments 2: [Figure 1 is quite "provisional." It doesn't add anything new, so it suggests removing or expanding upon the pathways in which this receptor participates.]
Response 2: Figure 1 has been expanded to include key molecular pathways. In addition, we have removed text from the illustration to make it more professional and visually coherent.
Comment 3: [However, I suggest improving the quality of Figure 2.]
Response 3: Figure 2 has been slightly reduced and optimised to improve its overall quality.
We are also grateful for your insightful question regarding our future research plans. This review is intended as a prelude to a larger, original study. The specific experimental model (cell lines or patient-derived material) will be selected depending on the availability of resources. Once again, thank you for your valuable comments and encouraging review.
Round 2
Reviewer 1 Report
Comments and Suggestions for Authors
No further comments
Reviewer 3 Report
Comments and Suggestions for Authors
I thank the authors for adding a couple of additional tables to the revised version of their manuscript in an effort to better summarize the current supportive evidence for P2X7R in gliomas. These additions nicely summarize the current P2X7R findings in a clear and succinct manner. Unfortunately, this also points out to the scarcity of data supporting the role of this receptor in glioma pathology.
I am still not convinced that the current evidence summarized in this manuscript for a role for P2X7R in gliomas is strong enough to support the conclusions drawn by the authors. Furthermore, the evidence for this receptor as a potential therapeutic target in gliomas is quite weak. For such conclusions, tumor cell proliferation/survival data conducted with glioma cells in which P2X7R is knocked out/knocked down corroborated with tumor growth studies in animals implanted intracranially with receptor knocked out/knocked down xenografts plus data from in vivo conditional knockout studies targeting various tumor microenvironment components in tumor-bearing animals would be the minimal evidence required for such qualifications. Such data are currently missing.
On the other hand, the role of Erbb2 as an oncogenic driver in Erbb2-amplified breast cancer, which is given as an example by the authors, is much better established and supported by numerous preclinical and clinical studies spanning several decades. By comparison, the evidence for P2X7R involvement in glioma pathology is not quite there yet.